# In Vitro Antiviral Activity of *Rhodiola crenulata* Extract against Zika Virus and Japanese Encephalitis Virus: Viral Binding and Stability

**DOI:** 10.3390/ph17080988

**Published:** 2024-07-26

**Authors:** Zheng-Zong Lai, I-Chuan Yen, Hao-Yuan Hung, Chen-Yang Hong, Chih-Wei Lai, Yen-Mei Lee

**Affiliations:** 1Graduate Institute of Medical Science, National Defense Medical Center, Taipei 114, Taiwan; laizengzong@gmail.com; 2Department and Graduate Institute of Pharmacology, National Defense Medical Center, Taipei 114, Taiwan; hyhung@mail.ndmctsgh.edu.tw (H.-Y.H.); zxc457659@gmail.com (C.-Y.H.); 3School of Pharmacy, National Defense Medical Center, Taipei 114, Taiwan; yenichuan@mail.ndmctsgh.edu.tw (I.-C.Y.); penghupharmacy@mail.ndmctsgh.edu.tw (C.-W.L.)

**Keywords:** antiviral agent, *Rhodiola crenulata*, Zika virus, Japanese encephalitis virus, salidroside, gallic acid

## Abstract

Zika virus (ZIKV) and Japanese encephalitis virus (JEV) can cause permanent neurological damage and death, yet no approved drugs exist for these infections. *Rhodiola crenulate,* an herb used in traditional Chinese medicine for its antioxidation and antifatigue properties, was studied for its antiviral activity against ZIKV and JEV in vitro. The cytotoxicity of *Rhodiola crenulata* extract (RCE) was evaluated using the CCK-8 reagent. Antiviral effects of RCE were assessed in ZIKV-infected or JEV-infected Vero cells via quantitative reverse transcription polymerase chain reaction (qRT-PCR), Western blotting, fluorescent focus assay (FFA), and immunofluorescence assay (IFA). The cell-free antiviral effects of RCE were evaluated using an inactivation assay. To determine the stage of the viral life cycle affected by RCE, time-of-addition, binding, and entry assays were conducted. Three bioactive constituents of RCE (salidroside, tyrosol, and gallic acid) were tested for antiviral activity. RCE exhibited dose-dependent anti-ZIKV and anti-JEV activities at non-cytotoxic concentrations, which were likely achieved by disrupting viral binding and stability. Gallic acid exhibited antiviral activity against ZIKV and JEV. Our findings indicate that RCE disrupts viral binding and stability, presenting a potential strategy to treat ZIKV and JEV infections.

## 1. Introduction

The outbreak of Zika virus (ZIKV), a flavivirus in the family Flaviviridae, in South America in 2015 and the subsequent increase in microcephaly in Brazil have raised significant health concerns. Native mosquito-borne ZIKV has been detected in 87 countries and territories [1], highlighting the urgent need for effective antiviral interventions. Approximately 80% of ZIKV-infected individuals are asymptomatic or present with only mild symptoms [2]. However, ZIKV infection in adults can lead to severe neurological conditions such as Guillain–Barré syndrome [3,4,5]. Moreover, ZIKV infection in pregnant women increases the risk of miscarriage or severe neurological defects in neonates, including microcephaly, intracranial calcification, and blindness. A study on ZIKV infection among pregnant women in the U.S. and its territories revealed that ZIKV-associated birth defects occurred in 4.6% of infants [6]. Research in pregnant mice has shown that ZIKV can cause placental damage and fetal death [7] and that the virus can invade the placenta by crossing or bypassing the placental barrier during pregnancy [8]. ZIKV is unique among flaviviruses for its ability to be transmitted by mosquitoes, parent-to-offspring transmission, and sexual contact [1,9].

Japanese encephalitis virus (JEV) is another positive-sense single-stranded RNA virus in the genus Flavivirus. JEV infections primarily occur in East and Southeast Asia during the rainy season [10]. JEV can spread between humans and animals in natural environments, with enzootic transmission cycles involving amplifier hosts such as swine, wading birds, and mosquitoes [11]. Despite the availability of the JEV vaccine since the mid-1900s, JEV remains a leading cause of viral encephalitis in endemic areas [12], with an estimated 70,000 cases annually, primarily in China and India [13]. Symptoms of JEV range from mild to severe and, in some cases, the disease can be fatal. Notably, 30–50% of survivors experience permanent neurological sequelae or disability [14]. Currently, no anti-ZIKV or anti-JEV drugs have been approved, limiting treatment efforts to symptomatic relief.

Research teams globally are striving to develop novel antiviral drugs or to repurpose existing drugs. For decades, traditional Chinese herbal medicines have been a rich source of novel compounds for drug development. However, most herbal extracts obtained using organic solvents or water contain various compounds, making it challenging to isolate the source of the observed pharmacological effects. Herbal plant extracts rich in polyphenols, flavonoids, and terpenoids are promising candidates for the development of drugs targeting ZIKV infection [15,16].

*Rhodiola crenulata* (Figure 1A), belonging to the *Rhodiola* genus of the Crassulaceae family, thrives in high-altitude areas above 4000 m, such as the Tibetan Plateau and Northern Europe [17]. It has been used in traditional Chinese herbal medicine for centuries, with its roots and rhizomes known for containing more than 160 identified chemical constituents [18]. These compounds encompass acyclic alcohol derivatives, benzyl and phenol derivatives, phenylethane derivatives, phenylpropanoid derivatives, flavonoid derivatives, gallic acid derivatives, and other compounds. Salidroside, tyrosol, and gallic acid, along with their derivatives, are notable constituents found in *Rhodiola* species [18].

Researchers have documented numerous beneficial effects of *Rhodiola* species, including anti-aging, antioxidant, anti-inflammatory, anti-cancer, anti-hypoxic, anti-fatigue, anti-altitude sickness, and antidiabetic effects [19,20,21,22,23]. In addition, *Rhodiola* possesses antiviral activity. For instance, *R. kirilowii (Regel) Maxim*. Inhibits hepatitis C virus (HCV) infection [24], *R. imbricata* inhibits dengue virus [25], and *R. rosea* inhibits Ebola virus, Marburg virus [26], and influenza viruses [27].

Given that DENV, ZIKV, and JEV belong to the same genus (Flavivirus), and HCV belongs to the family Flaviviridae, the current study aimed to evaluate the antiviral efficacy of *Rhodiola crenulata* extract (RCE) against ZIKV and JEV in vitro, as well as the underlying mechanisms involved. We also assessed the inhibitory effects of the three characteristic constituents of RCE (salidroside, tyrosol, and gallic acid) (Figure 1B–D) on ZIKV and JEV. Overall, RCE exhibited anti-ZIKV and anti-JEV activities by disrupting viral binding and stability, potentially involving gallic acid.

## 2. Results

### 2.1. RCE Presented Anti-ZIKV Effects

The anti-ZIKV effects of RCE were assessed by treating infected Vero cells (MOI = 0.01) with RCE at the indicated concentrations of 0, 20, 40, and 60 µg/mL in a CO_2_ incubator at 37 °C for 48 h. Intracellular viral RNA levels were measured using quantitative reverse transcription polymerase chain reaction (qRT-PCR). RCE concentrations exceeding 40 µg/mL significantly reduced ZIKV RNA levels, with treatment at 60 µg/mL resulting in a 78% decrease in ZIKV RNA expression levels (Figure 2A). The IC_50_ value of the RCE against ZIKV was 50.96 µg/mL. The fluorescent focus assay (FFA) results showed that RCE treatment significantly reduced the ZIKV progeny yield (Figure 2B). The IFA and Western blotting results revealed that RCE treatment suppressed ZIKV infection and viral protein expression in a dose-dependent manner (Figure 2C,D). The concentrations of RCE used in the above experiments were not cytotoxic based on the cell viability results (Appendix A). Overall, these results demonstrated that RCE inhibited ZIKV growth without any cytotoxic effects, as evidenced by decreased viral RNA levels, progeny yield, and infection ability.

### 2.2. RCE Inhibited JEV Infection

The effects of RCE on JEV infection inhibition were assessed by inoculating JEV-infected Vero cells (MOI = 0.02) with increasing concentrations of RCE at 37 °C for 48 h. This analysis included intracellular JEV RNA levels (qRT-PCR), viral progeny yield (FFA assays), NS3 viral protein expression (Western blot analysis), and virus infection ability (IFA). Treatment with RCE at concentrations of 20–60 µg/mL significantly reduced JEV RNA levels. The IC_50_ value of RCE against JEV was determined to be 6.62 µg/mL, with treatment at 60 µg/mL reducing viral RNA levels by up to 99% (Figure 3A). Additionally, RCE treatment at concentrations of 20–60 µg/mL significantly decreased viral progeny yields (Figure 3B). IFA and Western blot results further confirmed that increasing concentrations of RCE inhibited JEV infection and NS3 protein expression (Figure 3C,D).

### 2.3. RCE Inhibited ZIKV and JEV Growth in the Early Stage of Infection

To investigate the impact of RCE on the viral life cycle, time-of-addition assays were performed. RCE (60 µg/mL) was administered to virus-infected cells (ZIKV MOI = 0.5; JEV MOI = 0.5) following a 2 h viral absorption period. The samples were divided into four groups according to treatment strategy: pre-treatment, co-treatment, post-treatment, and full-treatment (Figure 4A). After incubation for 24 h, viral RNA levels were assessed using qRT-PCR.

Virologists have determined that a reduction in viral infection following the administration of antiviral drugs during the pretreatment phase suggests that the drugs have a direct effect on host cells. A decrease in viral infection after the administration of antiviral drugs during the co-treatment phase suggests that the drugs act on the surface of the virus and its receptors on host cells. A decrease in viral infection after the administration of antiviral drugs (i.e., during the post-treatment phase) suggests that antiviral activity is associated with post-entry events such as viral replication [28]. In this study, the full-treatment and virus-control groups were used as positive and negative controls, respectively.

ZIKV RNA expression in the co-treatment group was 82% lower than that in the viral control group. No statistically significant differences were detected between the ZIKV RNA levels in the virus group and those in the pre- or post-treatment groups (Figure 4B). Time-of-addition assays revealed similar inhibitory effects in the anti-JEV experiment (Figure 4C), suggesting that RCE treatment disturbed the viral life cycle at the binding and entry stages. Increasing the RCE concentration throughout the co-treatment stage confirmed the dose-dependent inhibitory effects of RCE on ZIKV and JEV RNA expression (Figure 4D,E). These results suggest that the inhibitory effects of RCE occur in the early stages of viral infection, rather than in the later stages. However, the design of time-of-addition assays involved a relatively long incubation period (24 h), during which multiple viral replications could occur. Binding and entry assays were performed to investigate this process.

### 2.4. RCE Suppressed ZIKV and JEV Infection by Inhibiting Viral Binding

Virologists have found that at cold temperatures (4 °C), viruses can bind to cell surfaces but cannot enter cells [28]. This is the principle used in binding assays. Cells were exposed to RCE (60 µg/mL) and the virus (MOI = 1) simultaneously at 4 °C for 1 h, followed by measuring viral RNA levels via qRT-PCR. RCE treatment reduced ZIKV RNA expression levels by 78% and JEV RNA levels by 41% (Figure 5A,B), indicating that RCE treatment interfered with ZIKV and JEV binding.

Entry assays were also conducted by infecting cells with the virus (ZIKV MOI = 1; JEV MOI = 1) at 4 °C for 1 h, followed by washing and incubation with or without RCE (60 µg/mL) at 37 °C for 1 h and then analysis using qRT-PCR. The entry assays revealed no inhibitory effects of RCE on ZIKV or JEV entry (Figure 5C,D), further confirming that the antiviral effects of RCE are associated with viral binding.

### 2.5. RCE Can Inactivate ZIKV and JEV

Cell-free inactivation assays were performed to determine whether the RCE directly affected virion stability. Stock solutions of ZIKV or JEV (2 × 10^7^ FFU) were incubated with RCE at various concentrations (0, 20, 40, and 60 µg/mL) at 37 °C for 2 h. To avoid the effects of residual RCE in subsequent viral quantification, the drug-virus mixture was diluted 100-fold, and viral titers were then determined using FFA. The addition of RCE at concentrations of 20 or 60 µg/mL led to a 1.7 log and 2.8 log decrease, respectively, in ZIKV titers (Figure 6A) and a 0.1 log and 2.1 log decrease, respectively, in JEV titers (Figure 6B). RCE disrupted ZIKV and JEV stability.

### 2.6. Effects of Salidroside, Tyrosol, and Gallic Acid on ZIKV and JEV

We also sought to identify the RCE constituents responsible for the antiviral activity against ZIKV and JEV focusing on salidroside, tyrosol, and gallic acid, which have been previously identified as characteristic compounds of *Rhodiola* species. The presence of these compounds in the RCE was validated by HPLC (Appendix A) and LC mass spectrometry (Appendix A). This analysis began with the assessment of cytotoxicity in Vero cells. Cell viability tests based on CCK-8 assays revealed the following half-maximal cytotoxic concentration (CC_50_) values: salidroside (>480 µg/mL), tyrosol (>353.3 µg/mL), and gallic acid (>108.8 µg/mL) (Appendix A). To avoid drug-induced cytotoxicity in antiviral experiments, the optimal concentrations of salidroside, tyrosol, and gallic acid were limited to 240 µg/mL, 88.3 µg/mL, and 27.2 µg/mL, respectively.

Viral RNA levels were measured via qRT-PCR after ZIKV-infected Vero or JEV-infected Vero cells were treated with salidroside, tyrosol, or gallic acid (MOI = 0.5) for 24 h. Salidroside and tyrosol had no discernible effect on ZIKV RNA levels (Figure 7A,B); however, gallic acid had a significant dose-dependent inhibitory effect (Figure 7C). Similarly, only gallic acid significantly reduced viral RNA levels in JEV-infected Vero cells (Figure 7D–F). The IFA results further confirmed that gallic acid decreased the infectious abilities of ZIKV and JEV in a dose-dependent manner (Figure 8A,B). These results suggest that gallic acid plays a crucial role in the antiviral effects of RCE against ZIKV and JEV.

## 3. Discussion

The neurological damage induced by ZIKV and JEV remains a significant priority for antiviral drug development and public health. Although JEV-infected human mothers giving birth to infants with microcephaly have not been reported, compelling evidence indicates the persistence of JEV in the endometrium and placenta following transplacental or fetal infections in a pig model [29]. For both ZIKV and JEV infections, any reduction in viremia can theoretically mitigate the adverse consequences of viral infection. Therefore, there is an urgent need to develop drugs capable of counteracting the detrimental effects of these diseases.

*Rhodiola crenulata* belongs to the genus *Rhodiola* of the family Crassulaceae. *Rhodiola* is known for its anti-fatigue, antioxidation, anti-aging, and anti-inflammatory effects [18]. For decades, food-grade *Rhodiola* extracts have been widely used in Chinese folk medicine, with no reported issues, indicating that *Rhodiola*-based products are safe supplements. Researchers have also reported the antiviral activity of *Rhodiola* against hepatitis C virus, dengue virus-2, Ebola virus, Marburgvirus, and influenza viruses. However, the antiviral effects of RCE against ZIKV and JEV have not been previously investigated.

In this study, we demonstrated the antiviral effects of RCE against ZIKV and JEV in vitro. The IC_50_ values were calculated using the Prism software application (Version 6.0) based on four concentrations (0, 20, 40, and 60 µg/mL). We recognized that this range of concentrations is limited, and that the IC_50_ obtained is a preliminary estimate. The primary aim of this study was to demonstrate the dose-dependent inhibitory effects of this extract. It is important to note that we initially included higher concentrations beyond 60 µg/mL in our experiments. As demonstrated in Appendix A, concentrations above 60 µg/mL significantly impacted cell viability. This observation necessitated the use of lower concentrations (0, 20, 40, and 60 µg/mL) for our IC_50_ calculations to ensure cell viability and obtain meaningful data. In the dose-dependent assays, RCE exhibited anti-ZIKV and anti-JEV effects based on viral RNA levels (Figure 2A and Figure 3A), progeny yields (Figure 2B and Figure 3B), viral infection ability (Figure 2C and Figure 3C), and viral protein expression levels (Figure 2D and Figure 3D). Notably, the inhibitory and antiviral effects of RCE were observed at concentrations exceeding 20 µg/mL. In addition, the inhibitory effects of RCE on JEV were more pronounced than those on ZIKV alone.

The inhibitory effect of RCE against ZIKV and JEV in the early stages of viral infection was observed in time-of-addition assays (Figure 4B–E). Further analysis using binding and entry assays revealed that the antiviral activity of RCE could be attributed to viral binding rather than viral entry (Figure 5A–D). Specifically, the RCE prevents the virus from binding to the cell surface. The binding process of flaviviruses is dependent on the envelope (E) protein, a critical structural component of viral particles. The viral E protein binds to attachment factors (e.g., such as glycosaminoglycans), resulting in high-affinity receptor binding [30], followed by clathrin-mediated endocytosis [31]. A C-type lectin receptor, DC-SIGN, mediates this interaction between ZIKV [32] and JEV [33]. However, we did not obtain direct evidence that the RCE interferes with the binding between DC-SIGN and the cell surface. Further research is needed to determine whether RCE disrupts the interactions between E protein, DC-SIGN, glycosaminoglycans, and other binding receptors.

Researchers have previously detected infectious ZIKV particles in the breast milk of ZIKV-infected mothers [34], raising concerns about vertical transmission. Antiviral drugs capable of inactivating viruses can alleviate this concern. In the present study, an analysis in a cell-free environment revealed that RCE decreased ZIKV and JEV stability in a dose-dependent manner (Figure 6A,B). Identifying the compound(s) responsible for the observed antiviral effects was challenging because of the presence of at least 160 constituent compounds in the RCE. Therefore, we focused our analysis on salidroside, tyrosol, and gallic acid, which are the characteristic constituents of *Rhodiola* species. Only gallic acid exhibited anti-ZIKV and anti-JEV activities in infected Vero cells (Figure 7C,F and Figure 8A,B). Gallic acid, a minor constituent of many plants, inhibits various viral infections, including human rhinovirus (HRV) [35], enterovirus 71 (EV-71) [36], human immunodeficiency virus [10,37], influenza A (H1N1) [38], and HCV [39], at different stages of the viral cycle. Our results broaden the antiviral range of gallic acid by demonstrating its anti-ZIKV and anti-JEV effects.

One of the primary limitations of our study is the absence of in vivo tests. Although our in vitro experiments provide valuable insights into the antiviral effects of *Rhodiola crenulata* extract and its active constituents, they do not fully capture the complexity of the biological interactions within a living organism. In vivo studies are essential to validate the efficacy and safety of this extract in a more physiologically relevant context. Future research should focus on conducting comprehensive in vivo experiments to corroborate our in vitro findings and explore the therapeutic potential of *Rhodiola crenulata* in treating viral infections. Additionally, in vivo studies will help to elucidate the pharmacokinetics, bioavailability, and potential side effects of the extract, thereby providing a more holistic understanding of its antiviral properties. Evaluating the potential of RCE in combination therapies, exploring its efficacy against a broader range of viruses, and investigating the underlying mechanisms are essential directions for future research.

## 4. Materials and Methods

### 4.1. Cells, Viruses, and Agents

RCE was kindly provided by Dr. Lee Shih-Yu (Graduate Institute of Aerospace and Undersea Medic, National Defense of Medical Center, Taiwan). The methods used for the authentication of *R. crenulata* (voucher specimen NDMCP no. 1000901) and the preparation of RCE were described in the previous literature [40]. RCE was dissolved in dimethyl sulfoxide (DMSO) to form a stock solution at a concentration of 50 mg/mL. Three standard compounds (salidroside, tyrosol and gallic acid) were purchased from ChemFaces (Wuhan, China) and were dissolved in DMSO to form stock solutions at concentrations of 40 mM. The three constituents (salidroside, tyrosol, and gallic acid) in RCE were validated by high-performance liquid chromatography (Appendix A). African green monkey kidney Vero cells (ATCC CRL-1586) were grown in Dulbecco’s modified Eagle’s medium (DMEM) with 5% fetal bovine serum in an incubator under 5% CO_2_ at 37 °C. The strains examined in this study were ZIKV (PRAVABC59) and JEV (RP9). The propagation and titration of ZIKV were conducted using Vero cells. The propagation and titration of JEV were performed using C6/36 and Vero cells, respectively. Viral titers were determined via fluorescence focus assay (FFA) as previously described [41].

### 4.2. Cytotoxicity Assay

The cytotoxicity of RCE and other compounds was assessed using a Cell Counting Kit 8 (CCK-8, Dojindo, Japan) in accordance with the manufacturer’s instructions. Briefly, the test compounds were administered at increasing concentrations to a cell monolayer in 96-well plates over a 48 h culture period. The supernatant was replaced with 100 μL of DMEM containing 10% CCK-8 reagent. After 1 h of incubation, the absorbance at 450 nm was measured using a spectrophotometer (Synergy HT, Biotek, Winooski, VT, United States). Cell viability in the compound groups was normalized to that in the vehicle group. Note that the content of DMSO in each group was adjusted to the same level to prevent interference from the solvent [41]. Note also that these assays were performed in triplicate.

### 4.3. Quantitative Reverse Transcription-Polymerase Chain Reaction (qRT-PCR)

Intracellular viral RNA levels were measured using qRT-PCR. Briefly, total cellular RNA was extracted using TRIzol reagent (Invitrogen, Tokyo, Japanese) to measure viral RNA levels using the One-Step 2X qRT-PCRmix SYBR Green Kit (Bioman, QRP001, New Taipei City, Taiwan). Table 1 lists the sequences of primer used to quantify the production of the ZIKV envelope, NS2 of JEV, and β-actin (as an internal control). qRT-PCR was performed using a Roche Lightcycler 480 instrument (Roche Applied Science, Indianapolis, Indiana) via the following steps: reverse transcription at 42 °C for 20 min and initial denaturation at 95 °C for 10 min, followed by 40 cycles at 95 °C for 10 s, 62 °C for 15 s, and 72 °C for 20 min. Each assay was performed in triplicate, and the data were calculated using the 2^−ΔΔCt^ method. The viral RNA levels in the drug groups were normalized to those in the vehicle group [41].

### 4.4. Fluorescent Focus Assay (FFA) and Immunofluorescence Assay (IFA)

FFA is a time-saving method for virus quantification based on immunostaining. In the present study, tenfold serially diluted virus solutions were added to a cell monolayer. After a 2 h infection period, the medium was discarded, and the cells were washed twice with phosphate-buffered saline (PBS). Semisolid DMEM supplemented with 1.2% methylcellulose was then added to the cells for a 48 h incubation, after which an indirect immunofluorescence assay (IFA) was performed. The cells were fixed with 4% formaldehyde for 30 min, after which permeabilization was performed using a 1:1 mixture of methanol/acetone. After incubation with blocking buffer (5% skim milk) for 2 h, the cells were washed twice with PBS. The cells were incubated at 25 °C for 2 h with primary antibodies for the detection of ZIKV Env protein, including 4G2 antibodies or anti-JEV NS3 protein antibodies (proprietary). The cells were then washed again with PBS prior to a 2 h incubation with Alexa Fluor 488-conjugated secondary antibodies (goat anti-mouse IgG). Finally, the cells were washed with PBS before viral infection was observed under a fluorescence microscope. Viral titers were determined and expressed as fluorescence foci units per milliliter (FFU/mL) [41].

### 4.5. Western Blot Analysis

The treated cells were harvested and lysed using RIPA buffer (Bioman, Scientific Co., Ltd. New Taipei City, Taiwan). Cell debris was removed via centrifugation at 10,000× *g* at 4 °C for 15 min. Protein separation was performed using sodium dodecyl sulfate-polyacrylamide gel electrophoresis, after which the proteins were transferred to a PVDF membrane via a semidry transfer system (Bio-Rad, Hercules, CA, USA). The membrane was blocked using blocking buffer at room temperature for 2 h, after which the primary antibodies used for the detection of ZIKV Env protein or anti-JEV NS3 protein (proprietary) were added to the cells and incubated at 25 °C for 2 h. Horseradish peroxidase-conjugated goat anti-rat secondary antibodies (1:4000) were added to the membranes after removing the primary antibodies, after which the membranes were washed three times with TBST. After incubation for 2 h, the membranes were washed again with TBST before an enhanced chemiluminescent reagent was added. Bands were detected using a UVP chemiluminescence detection system.

### 4.6. Time-of-Addition Assay

RCE (60 µg/mL) was administered to cells (ZIKV MOI = 0.5; JEV MOI = 0.5) relative to a 2 h absorption period using four treatment strategies: pre-treatment (pre; administration of drug at 1 h prior to viral infection), co-treatment (co; co-administration of drug and virus for 2 h absorption), post-treatment (post; administration of drug after 2 h viral absorption period), and full-treatment (full; administration of drug from the beginning of virus infection to the end of 24 h incubation). Viral RNA levels were assessed after 24 h using qRT-PCR, and all experiments were performed in triplicate.

### 4.7. Binding Assay

Cells were infected with a virus (ZIKV MOI = 1; JEV MOI = 1) with or without 60 µg/mL RCE in 12-well plates and incubated at 4 °C for 1 h prior to analysis via qRT-PCR.

### 4.8. Entry Assay

Cells were infected with a virus (ZIKV MOI = 1; JEV MOI = 1) in 12-well plates at 4 °C for 1 h, followed by two washes with PBS to remove unabsorbed viruses. The cells were then incubated with or without 60 µg/mL RCE at 37 °C for 1 h prior to analysis using qRT-PCR.

### 4.9. Inactivation Assay

In a cell-free model, various concentrations of RCE were added to the virus solution and held at 37 °C for 2 h. Titers of the drug-treated virus solutions were determined using FFA. All experiments were conducted in triplicate.

### 4.10. Statistical Analysis

The results are presented as the mean ± standard deviation from assays performed in triplicate. Statistically significant differences were assessed using one-way ANOVA with Tukey’s multiple comparisons test. A *p*-value less than 0.05 indicated statistical significance. The GraphPad Prism 6.0 software application was used for all the statistical analyses.

## 5. Conclusions

Taken together, these results indicate that RCE is an effective antiviral agent against ZIKV and JEV that reduces viral attachment and stability. Our findings also implicated gallic acid in these effects. Furthermore, our study provides evidence that RCE is a promising candidate for the development of ZIKV and JEV treatments.

## Figures and Tables

**Figure 1 pharmaceuticals-17-00988-f001:**
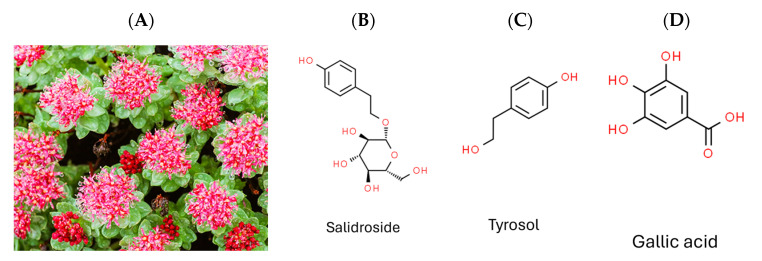
Photograph of plant *Rhodiola crenulata* and chemical structures of target compounds. (**A**) *Rhodiola crenulata*. (**B**) Salidroside. (**C**) Tyrosol. (**D**) Gallic acid.

**Figure 2 pharmaceuticals-17-00988-f002:**
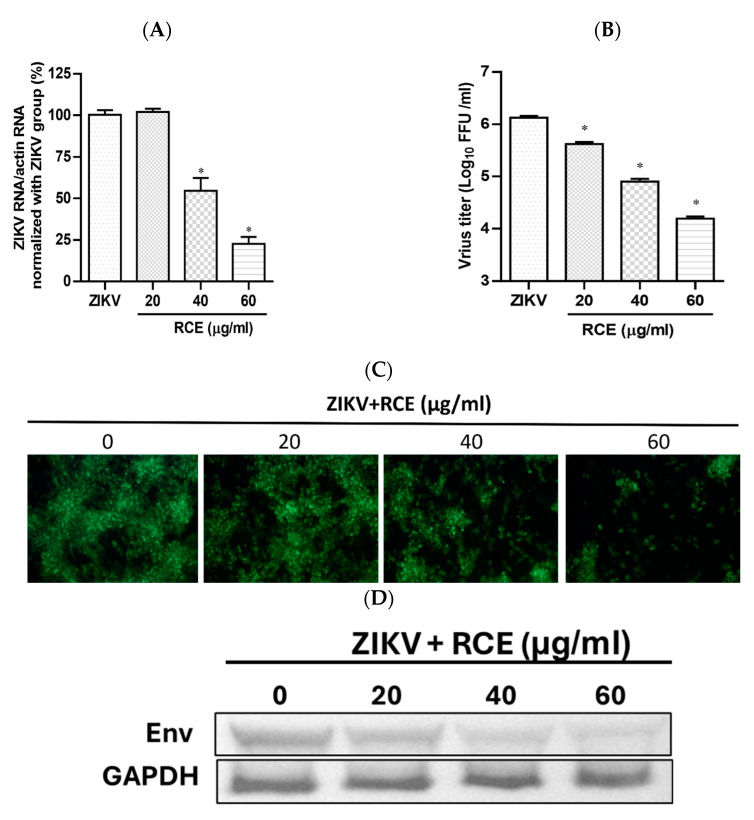
Antiviral activity of *Rhodiola crenulata* extract (RCE) on Zika virus (ZIKV). Infected Vero cells (MOI = 0.01) were incubated in 12-well plates with various concentrations of RCE for 48 h: (**A**) Intracellular ZIKV RNA levels were determined using qRT-PCR. (**B**) Extracellular progeny yields were determined using fluorescent focus assay (FFA). (**C**) The inhibitory effects on the infection ability were determined using immunofluorescence assay (IFA). Magnification power of the microscope was 200 times. Green fluorescence indicates virus-infected cells. (**D**) ZIKV Env protein expression was measured using Western blot analysis. The data are presented as the mean ± SD of experiments conducted in triplicate. Statistical significance was assessed via one-way ANOVA versus the virus control group: * indicates *p* < 0.05.

**Figure 3 pharmaceuticals-17-00988-f003:**
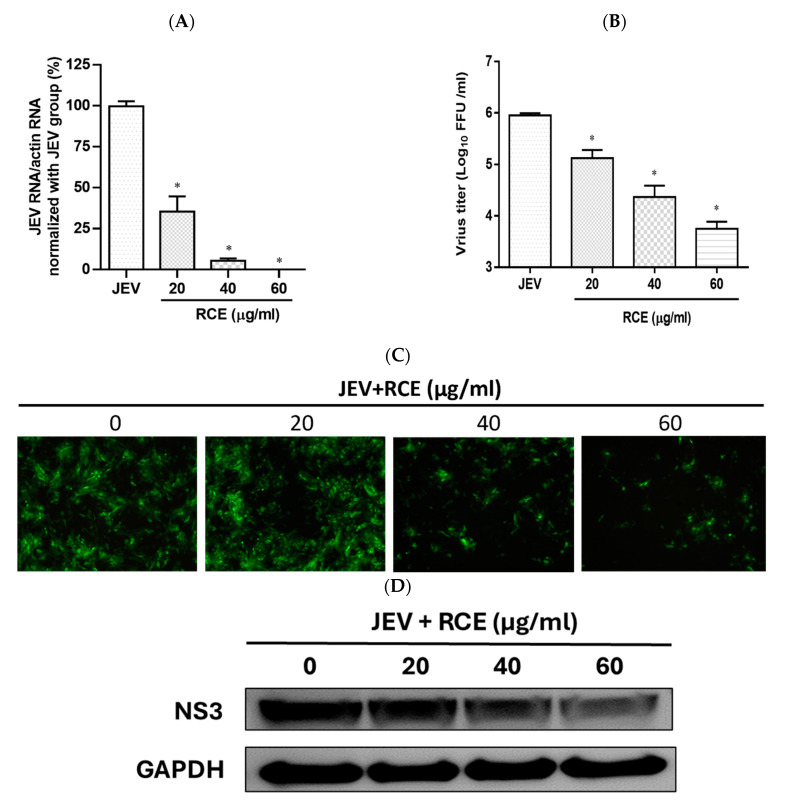
Antiviral activity of *Rhodiola crenulata* extract (RCE) on Japanese encephalitis virus (JEV). Vero cells were inoculated with JEV (MOI = 0.02) in 12-well plates with indicated concentrations of RCE for 48 h: (**A**) Intracellular JEV RNA levels were determined using qRT-PCR. (**B**) Extracellular progeny yields were determined using fluorescent focus assay (FFA). (**C**) The inhibitory effects on infection were determined using immunofluorescence assay (IFA). Magnification power of the microscope was 200 times. Green fluorescence indicates virus-infected cells. (**D**) JEV NS3 protein expression was determined via Western blot analysis. The data are presented as the mean ± SD of experiments conducted in triplicate. Statistical significance was assessed via one-way ANOVA versus the virus control group: * indicates *p* < 0.05.

**Figure 4 pharmaceuticals-17-00988-f004:**
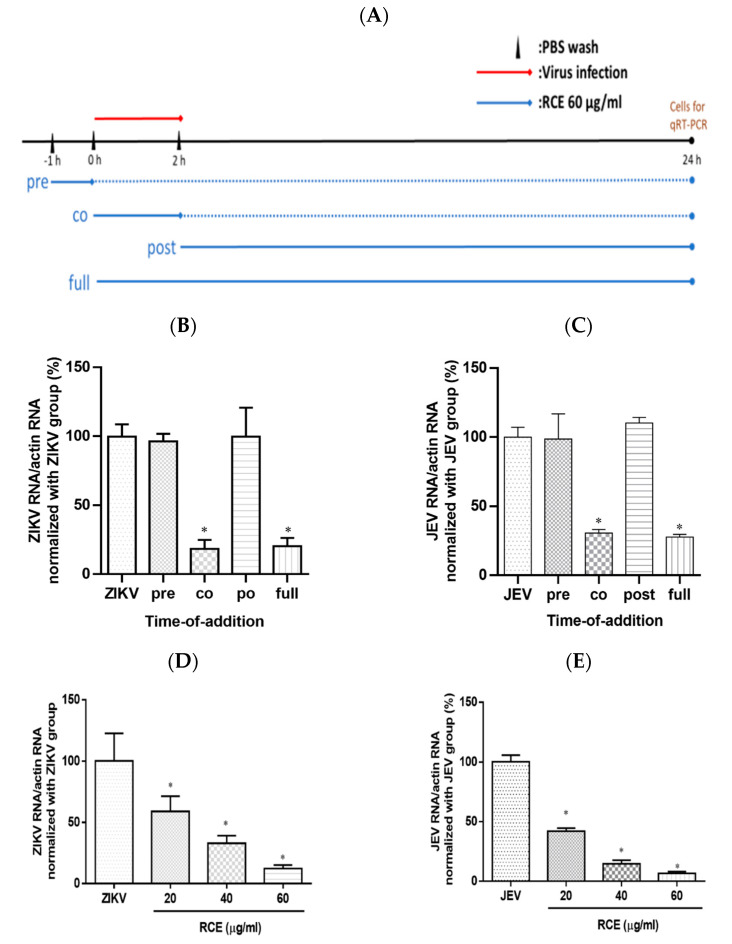
Time-of-addition assay results. (**A**) Timeline of time-of-addition assays. The red line indicates the viral infection, the blue line indicates the *Rhodiola crenulata* extract (RCE) administration period, and the dotted line represents the incubation period. Infected cells (MOI = 0.5) were treated with RCE in 12-well plates at various time points relative to viral infection. After a 24 h incubation. (**B**) Zika virus (ZIKV) or (**C**) Japanese encephalitis virus (JEV) RNA levels were measured using qRT-PCR. Dose-inhibition assays involved the simultaneous exposure of cells to the drug and virus at increasing concentrations (MOI = 0.5) for 2 h (absorption period), after which the supernatant was discarded and the cells were washed twice with PBS. (**D**) ZIKV or (**E**) JEV RNA expression was measured using qRT-PCR after incubation for 24 h. The data are presented as the mean ± SD of experiments conducted in triplicate. Statistical significance was assessed via one-way ANOVA versus the virus control group: * indicates *p* < 0.05.

**Figure 5 pharmaceuticals-17-00988-f005:**
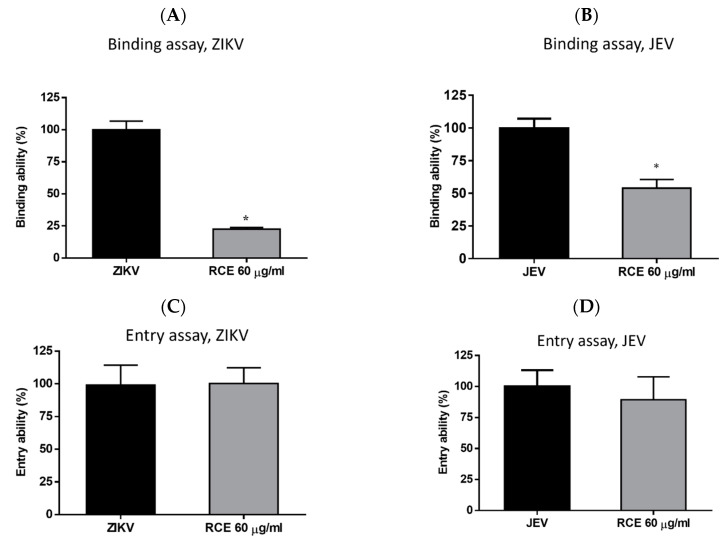
Binding assays for (**A**) Zika virus (ZIKV) and (**B**) Japanese encephalitis virus (JEV). Entry assays for (**C**) ZIKV and (**D**) JEV. The data are presented as the mean ± SD of experiments conducted in triplicate. Statistical significance was assessed via one-way ANOVA versus the virus control group: * indicates *p* < 0.05.

**Figure 6 pharmaceuticals-17-00988-f006:**
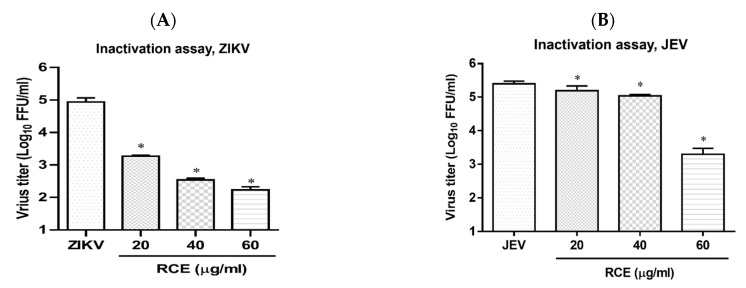
Inactivation assay of *Rhodiola crenulata* extract (RCE) on Zika virus (ZIKV) and Japanese encephalitis virus (JEV), which involved incubating RCE with (**A**) ZIKV (2 × 10^7^ FFU) or (**B**) JEV (2 × 10^7^ FFU) at 37 °C for 2 h. Virion stability was evaluated using FFA. The data are presented as the mean ± SD of experiments conducted in triplicate. Statistical significance was assessed via one-way ANOVA versus the virus control group: * indicates *p* < 0.05.

**Figure 7 pharmaceuticals-17-00988-f007:**
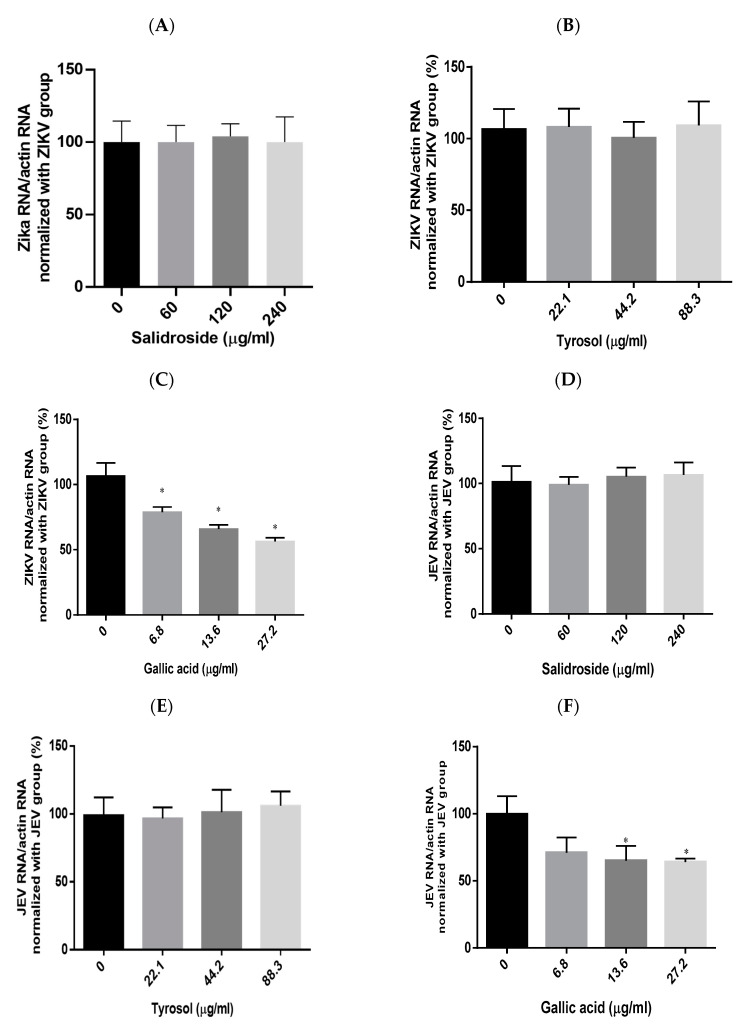
Antiviral effects of *Rhodiola crenulata* extract (RCE) constituents, salidroside, tyrosol, and gallic acid on (**A**–**C**) Zika virus (ZIKV) and (**D**–**F**) Japanese encephalitis virus (JEV). After treating virus-infected Vero cells (ZIKV MOI = 0.01; JEV MOI = 0.02) with each of the constituent compounds for 24 h, viral RNA levels were measured using qRT-PCR. Statistical significance was assessed via one-way ANOVA versus the virus control group: * indicates *p* < 0.05.

**Figure 8 pharmaceuticals-17-00988-f008:**
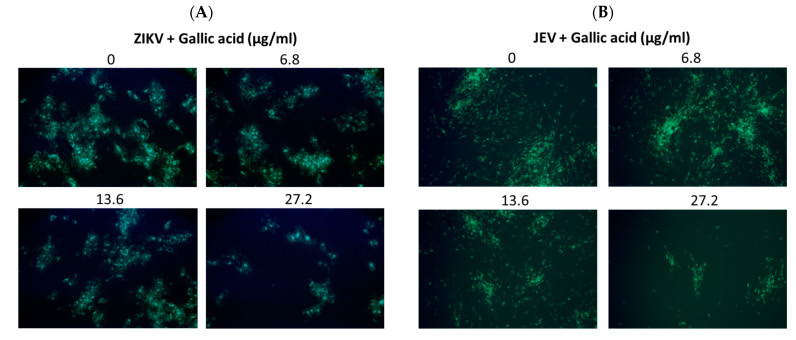
Antiviral activity of gallic acid on Zika virus (ZIKV) and Japanese encephalitis virus (JEV). Infected Vero cells were inoculated in 12-well plates with indicated concentrations of gallic acid for 48 h. Magnification power of the microscope was 200 times. The inhibitory effects on infection were determined using immunofluorescence assay (IFA). Green fluorescence indicates virus-infected cells. IFA results for (**A**) ZIKV and (**B**) JEV.

**Table 1 pharmaceuticals-17-00988-t001:** Primer sequences for qRT-PCR.

Name	Orientation	Primer Sequence
		5′-3′Orientation
ZIKA Env	Forward	TTGGTCATGATACTGCTGATTGC
	Reverse	CCTTCCACAAAGTCCCTATTGC
JEV NS2	Forward	TCCGTCACCATGCCAGTCTT
	Reverse	GAGGATGATTCTGTAAGTATC
β-actin	Forward	AGGCACCAGGGCGTGAT
	Reverse	GCCCACATAGGAATCCTTCTG

## Data Availability

The datasets used in the current study are available from the corresponding author upon reasonable request.

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
