# Peer review of "In Vitro Antiviral Activity of Rhodiola crenulata Extract against Zika Virus and Japanese Encephalitis Virus: Viral Binding and Stability"

_pharmaceuticals, 2024, doi:10.3390/ph17080988_

Round 1

Reviewer 1 Report

Comments and Suggestions for Authors

Authors of the manuscript entitled In Vitro Antiviral Activity of Rhodiola Crenulata Extract against Zika Virus and Japanese Encephalitis Virus: Viral Binding and Stability investigated the antiviral activity of Rhodiola crenulata extract (RCE) against ZIKV and JEV in vitro.

But some points required more clarification by the authors.

1-In sections 2.1 & 2.2. Authors annotated for the IC50 values of the Rhodiola crenulata extract based on three different concentrations 20, 40, & 60 µg/mL. Obtaining IC50 values from just three point concentrations is inappropriate and this need further clarifications from the authors.

2-In figures 1C & 2C & 7, scale bars or magnification power should be annotated on the IFA images or at figure legends.

3-In section 2.3. Second paragraph requires a proper citation from reported studies and/or tertiary references, since it highlights important data interpretation.

4-In section 2.4, authors should rationalize why they adopted the 60 µg/mL for RCE application rather than the respective IC50 values for each virus obtained from earlier sections.

5-Again in section 2.4, the principle of binding assay requires proper referencing.

6-Authors are asked to elaborate more on the discussion section highlighting study limitations and future work.

7-In the text, reference numbers should be placed in square brackets [ ], in accordance with the requirements of MDPI style.

8-All scientific names should written in italic within the manuscript, for example the name in captions of figure 1 and 2, Abbreviations (line 402) as well as in references section.

9-what the source of the 3 compounds (salidroside, tyrosol, and gallic acid) analyzed in the study.

10-How the authors obtained the Rhodiola Crenulata extract.

Reviewer 2 Report

Comments and Suggestions for Authors

Medicinal plants have been used worldwide for centuries for health promoting as well as for the prevention and treatment of diseases due to active phytochemical components. The recognition of their clinical and pharmaceutical value is constantly increasing, although this varies between countries. Natural chemicals contained in medicinal plants are prescribed in traditional medicine as herbal teas or phytopharmaceuticals without isolation of active compounds. Many people around the world, especially from developing countries, who have limited access to allopathic medicines for economic reasons, still rely on herbal medicines as their main source of health and sometimes the only source of care. However, the use of medicinal plants is not limited to developing countries, as phytotherapeutic products are increasingly used in all countries recently.

In this context the present manuscript, as well as the previous studies made on Rodiola crenulata, bring useful information about this plant and its medicinal potential. By publishing this information, the authors bring this plant to the attention of scientists and it could increase the interest for a more in-depth study and the possibility of its use in medicine as an antiviral natural product.

The manuscript is very well written. The English version, the experiment methods and the results are clear and understandable. Identifying the compounds responsible for the antiviral effect brings value to this study.

Reviewer 3 Report

Comments and Suggestions for Authors

The manuscript seems interesting in its findings. Please consider these comments:

- The authors wrote << To identify the RCE constituents responsible for antiviral activity, three bioactive constituents (salidroside, tyrosol and gallic acid) of RCE were evaluated in infected Vero cells.>>. The reason for selecting those 3 compounds is not clear. Are these compounds major in the extract? if so, please quantify. 

- Please add a colored photo of the studied plant in the manuscript. 

- What is the positive control the authors used? it seems that there is no positive control in all experiments.

- The Latin plant name (Rhodiola crenulata), should be italicized in the whole manuscript.

- There is no information about the source of the plant, who confirmed its identification, and from which source it is obtained. 

Comments on the Quality of English Language

 Moderate editing of English language required

Reviewer 4 Report

Comments and Suggestions for Authors

The paper: "In Vitro Antiviral Activity of Rhodiola Crenulata Extract against Zika Virus and Japanese Encephalitis Virus: Viral Binding and Stability" has been reviewed.

The paper describes the evaluation of the antiviral activity and mechanism of action of Rhodiola Crenulata extract against two positive-stranded RNA viruses, JEV and ZIKV.

The paper is of interest, however before proceeding for publication, the following aspects need to be addressed:

The captions should be better described, by indicating the abbreviations used;

The identification of compounds is based on HPLC retention times, I suggest confirming the identity using the addition method. 

The cytotoxicity values and the antiviral activities of active substances should be reported in ug/mL to easily compare the obtained results with the whole extract.

The quantification of the main active substances should be performed, to compare the obtained activity values.

The limitations of the study, in particular the absence of in vivo tests should be stated in conclusions.

The quality of the figures should be improved.

Comments on the Quality of English Language

The manuscript should be checked to revise typos

Round 2

Reviewer 1 Report

Comments and Suggestions for Authors

The authors have vastly improved the manuscript based on reviewer comments.